# Understanding the Gastrointestinal Behavior of the Coffee Pulp Phenolic Compounds under Simulated Conditions

**DOI:** 10.3390/antiox11091818

**Published:** 2022-09-15

**Authors:** Silvia Cañas, Miguel Rebollo-Hernanz, Cheyenne Braojos, Vanesa Benítez, Rebeca Ferreras-Charro, Montserrat Dueñas, Yolanda Aguilera, María A. Martín-Cabrejas

**Affiliations:** 1Department of Agricultural Chemistry and Food Science, Faculty of Science, Universidad Autónoma de Madrid, C/ Francisco Tomás y Valiente 7, 28049 Madrid, Spain; 2Institute of Food Science Research (CIAL, UAM-CSIC), Universidad Autónoma de Madrid, C/ Nicolás Cabrera 9, 28049 Madrid, Spain; 3Grupo de Investigación en Polifenoles, Unidad de Nutrición y Bromatología, Facultad de Farmacia, Universidad de Salamanca, Campus Miguel de Unamuno, 37007 Salamanca, Spain

**Keywords:** coffee pulp, coffee by-products, phenolic compounds, caffeine, in vitro digestion, bioaccessibility, bioavailability, absorption, colonic biotransformation, phenolic metabolites

## Abstract

Numerous residues, such as the coffee pulp, are generated throughout coffee processing. This by-product is a source of antioxidant phytochemicals, including phenolic compounds and caffeine. However, the antioxidant properties of the phenolic compounds from the coffee pulp are physiologically limited to their bioaccessibility, bioavailability, and biotransformation occurring during gastrointestinal digestion. Hence, this study explored the phenolic and caffeine profile in the coffee pulp flour (CPF) and extract (CPE), their intestinal bioaccessibility through in vitro digestion, and their potential bioavailability and colonic metabolism using in silico models. The CPE exhibited a higher concentration of phenolic compounds than the CPF, mainly phenolic acids (protocatechuic, chlorogenic, and gallic acids), followed by flavonoids, particularly quercetin derivatives. Caffeine was found in higher concentrations than phenolic compounds. The antioxidant capacity was increased throughout the digestive process. The coffee pulp matrix influenced phytochemicals’ behavior during gastrointestinal digestion. Whereas individual phenolic compounds generally decreased during digestion, caffeine remained stable. Then, phenolic acids and caffeine were highly bioaccessible, while flavonoids were mainly degraded. As a result, caffeine and protocatechuic acid were the main compounds absorbed in the intestine after digestion. Non-absorbed phenolic compounds might undergo colonic biotransformation yielding small and potentially more adsorbable phenolic metabolites. These results contribute to establishing the coffee pulp as an antioxidant food ingredient since it contains bioaccessible and potentially bioavailable phytochemicals with potential health-promoting properties.

## 1. Introduction

The coffee industry produces 9 million tons of coffee every year, resulting in substantial amounts of by-products derived from the different stages of coffee processing [1]. The coffee pulp (CP) is the main by-product discarded during wet coffee processing, representing 29% of the dry weight of the whole coffee cherry [2]. The CP is mainly composed of carbohydrates (57.2%), fiber (16.2%), proteins (13.4%), and lipids (1.6%). In addition, the CP also contains antioxidant phytochemicals, such as phenolic compounds and methylxanthines (caffeine) [3]. The primary phenolic compounds in the CP are chlorogenic, protocatechuic, ferulic, caffeic, *p*-coumaric, and gallic acids, although it may also contain procyanidins [4]. However, the profile of phenolic compounds in the CP has not been extensively studied. Consistent studies have related the intake of dietary phenolic compounds with a decreased risk of chronic diseases associated with oxidative stress, such as cardiovascular diseases, cancer, obesity, and diabetes. Phenolic compounds can reverse oxidative stress due to their high antioxidant capacity and ability to regulate cellular processes [5]. Our research group has previously developed eco-friendly extraction methods for recovering phenolic compounds from the coffee husk (the CP’s counterpart from the dry method) and the coffee parchment using water-based extractions [6,7]. Likewise, Rebollo-Hernanz et al. [8,9] demonstrated the impact of phenolic compounds from coffee by-products on preventing inflammatory responses in macrophages, modulating insulin signaling and adipogenesis in adipocytes, and regulating hepatic glucose, lipid, and mitochondrial energy metabolism.

Nonetheless, a diet supplemented with the phenolic compounds from the CP does not guarantee their ascribed effects due to their limited bioavailability. Phenolic compounds’ absorption depends on their chemical structure and interactions with the food matrix [10]. Phenolic compounds are released and transformed throughout the digestion in the gastrointestinal tract and are mainly absorbed in the small and large intestines [11]. Simulated gastrointestinal in vitro digestion models are essential to elucidate the bioaccessibility of bioactive compounds. Although showing limitations compared to in vivo assays, in vitro models are faster, cheaper, easier, and ethically unrestricted approaches to simulating gastrointestinal digestions [12]. In vitro digestions coupled to cellular models can be applied to analyze the absorption and bioavailability of bioactive compounds [13]. Caco-2 cells represent a suitable intestinal absorption model because of their morphological similarity to the small intestine [14]. Phenolic compounds not absorbed in the small intestine reach the colon, where the microbiota metabolize them into lower molecular weight metabolites. Exploring the effect of colonic fermentation on phenolic compounds is essential since phenolic metabolites may be the absorbed form or can exert their biological activity in the colon [15]. However, both Caco-2 cell culture and colonic fermentation models entail limitations and disadvantages, such as long experiment times and associated costs. Thus, computational predictions are a valuable alternative to assessing the potential intestinal permeability of target compounds and simulating the phenolic compounds’ behavior on gut microbiota rapidly and inexpensively [16]. Studying how gastrointestinal digestion affects the bioaccessibility, bioavailability, and bioactivity of phenolic compounds and caffeine is essential for understanding the potential of the CP for food, nutrition, and health purposes. We hypothesized that the phytochemicals (phenolic compounds and caffeine) present in the flour and aqueous extract from the CP might be differently released, transformed, and absorbed during gastrointestinal digestion. Our objective was to evaluate the bioaccessibility of phenolic compounds and caffeine from the CP using a standardized in vitro simulated gastrointestinal digestion model and to predict bioactive compounds’ bioavailability and colonic biotransformation through in silico models.

## 2. Materials and Methods

### 2.1. Materials

Methanol, sodium hydroxide, formic acid, acetonitrile, hydrochloride acid, sodium carbonate, ferric chloride hexahydrate, potassium chloride, sodium bicarbonate, sodium chloride, and Folin–Ciocalteu reagent were supplied by Panreac Química SLU (Barcelona, Spain). Pure phytochemical standards (≥96%), including gallic, caffeic, chlorogenic, *p*-coumaric, protocatechuic, and vanillic acids, rutin and quercetin, and caffeine, were purchased from Sigma-Aldrich (Sigma-Aldrich, St. Louis, MO, USA), and extrasynthese (Genay, France). 6-Hydroxy-2,5,7,8-tetramethylchromane-2-carboxylic acid (Trolox), 2,2′-Azino-bis(3-ethylbenzothiazoline-6-sulfonic acid) (ABTS), potassium persulfate, 2,4,6-tris(2-pyridyl)-s-triazine (TPTZ), ammonium carbonate, calcium chloride dihydrate, magnesium chloride hexahydrate, potassium phosphate monobasic, porcine pepsin, porcine pancreatin, pronase E, and Viscozyme ® were purchased from Sigma-Aldrich (Sigma-Aldrich, St. Louis, MO, USA).

### 2.2. Sample Preparation

The CP was separated from *Coffea arabica* L. cherries by the wet processing and provided by “*Las Morenitas*” (Nicaragua). The CP was ground, yielding the CP flour (CPF). The flour was maintained in sealed flasks at −20 °C until use. The CP aqueous extract (CPE) was prepared according to extraction conditions previously optimized for the coffee husk [7]. Briefly, the CPF (0.02 g mL^−1^ solid-to-solvent ratio) was added to boiling water (100 °C) and mixed for 90 min. After extraction, the aqueous extract was filtered, frozen at −20 °C, freeze-dried, and stored at −20 °C until further usage.

### 2.3. In Vitro Simulated Gastrointestinal Digestion

The in vitro gastrointestinal digestion was carried out according to the INFOGEST in vitro digestion protocol with slight modifications [12]. Briefly, 1 g of the CPF or 100 mg of the CPE were mixed with the simulated salivary fluid for simulating the oral phase, and the mixture was maintained for 2 min at 37 °C under agitation. Amylase was not added due to the absence of starch in the samples [17]. The gastric phase was performed by combining the oral phase with simulated gastric fluid and adding porcine pepsin solution (2000 U mL^−1^ of digestion), and the samples were incubated at 37 °C for 2 h under stirring. The intestinal phase was simulated by mixing the gastric phase with simulated intestinal fluid containing pancreatin (100 U trypsin activity mL^−1^ of digest). The mixture was incubated at 37 °C while stirring for 2 h. In vitro colonic digestion was simulated following the protocol described by Papillo et al. [18]. Pronase E (5 mL, 1 mg mL^−1^) was added to the intestinal phase, and the pH was previously adjusted to 8.0. The samples were incubated at 37 °C for 1 h under stirring. After incubation, the samples were adjusted to pH 4, and 150 µL of Viscozyme was added. The samples were incubated at 37 °C for 16 h under agitation. A digestion blank was prepared for each digestive phase containing the enzymes and simulated digestion fluids. The supernatants and residues from each digestive stage were freeze-dried and stored at −20 °C until utilization.

### 2.4. Extraction of Free and Bound Phenolic Compounds

Free and bound phenolic compounds were extracted from the CPF following the protocol of Rebollo-Hernanz et al. [19]. For free phenolic compounds extraction, 1 g of the CPF and the residue obtained from each digestive phase of the CPF digestions were combined with 50 mL of methanol: H_2_O: HCl (80:19.9:0.1, *v*/*v*). The samples were sonicated for 30 min in an ultrasonic bath and incubated at 40 °C for 16 h under agitation. After incubation, the samples were centrifuged at 4000× *g* (Centrifuge 5804R, Eppendorf, Hamburg, Germany) for 15 min at room temperature, separating the obtained supernatants. The extraction process was repeated twice. The supernatants collected from both extractions were combined and then concentrated at 40 °C under vacuum for methanol evaporation using a rotary evaporator (Rotavapor R-124, Buchi, Flawil, Switzerland). Alkaline extraction of bound phenolic compounds was conducted by adding 5 mL of 4 mol L^−1^ NaOH to the residue obtained after free phenolic compounds extraction. The samples were then agitated for 1 h under an N_2_ atmosphere at room temperature. The samples were acidified to pH 2.0 and centrifuged at 4000× *g* (Centrifuge 5804R, Eppendorf, Hamburg, Germany) for 15 min at room temperature, collecting the organic fraction. Afterward, extraction with ethyl ether 1:1: ethyl acetate was performed. Three more extractions were performed using methanol: H_2_O: HCl (80:19.9:0.1, *v*/*v*). Finally, the supernatants recovered from each extraction were combined and concentrated at 40 °C under vacuum (Rotavapor R-124, Buchi, Flawil, Switzerland) for solvent evaporation.

### 2.5. Spectrophotometric Assessment of the Total Phenolic Content and Antioxidant Capacity

#### 2.5.1. Total Phenolic Content (TPC)

The TPC was assessed using the Folin–Ciocalteu method [20], according to a previously adapted protocol [21]. Briefly, 10 μL of the sample was added to each well in a 96-well plate. Subsequently, 150 μL of diluted Folin–Ciocalteu reagent (1:14, *v*/*v* in Milli-Q water) were pipetted into each well, and the plate was incubated at room temperature for 3 min. Finally, 50 μL of Na_2_CO_3_ 20% was added to the mixture, and the plate was incubated for 2 h at room temperature in the dark. After incubation, absorbance was recorded at 750 nm in a microplate reader (BioTek Cytation 5, Biotek, Winooski, VT, USA). A standard curve of gallic acid (0.01–0.2 mg mL^−1^) was performed to estimate the concentration of total phenolic compounds. The results were expressed as mg gallic acid equivalents per gram (mg GAE g^−1^).

#### 2.5.2. ABTS Radical Scavenging Capacity

The radical scavenging capacity was measured by the ABTS assay as previously described [21]. To obtain 2.2′-azino-bis(3-ethylbenzothiazoline-6-sulfonic) acid radical cations (ABTS^•+^), a solution of ABTS (7 mmol L^−1^) was reacted with potassium persulfate (2.45 mmol L^−1^). The mixture was shaken for 16 h (room temperature, dark conditions). The ABTS^•+^ stock solution was adjusted to an absorbance of 0.70 at 734 nm by diluting with PBS (5 mmol L^−1^, pH 7.4). The assay was performed by mixing the sample (30 μL) with the ABTS^•+^ solution (270 μL) in a 96-well plate. After incubating for 10 min, the absorbance was measured at 734 nm in a microplate reader (BioTek Cytation 5, Biotek, Winooski, VT, USA). Trolox was used as the standard to prepare a calibration curve (0–0.06 mg mL^−1^). The results were transformed to mg Trolox equivalents per gram (mg TE g^−1^).

#### 2.5.3. Ferric Reducing Antioxidant Power (FRAP)

The antioxidant capacity was measured through the FRAP assay as previously described [19]. Samples (10 μL) were mixed with 300 μL of a working FRAP reagent (acetate buffer 0.3 mol L^−1^ pH 3.6, 10 mmol L^−1^ tripyridyl s-triazine, 40 mmol L^−1^ HCl, 20 mmol L^−1^ FeCl_3_·6H_2_O (10:1:1) (*v*/*v*/*v*)) in a 96-well plate. After incubating the plate for 10 min at 37 °C, the absorbance was read at 593 nm in a microplate reader (BioTek Cytation 5, Biotek, Winooski, VT, USA). Trolox was used as a standard solution (25–800 µmol L^−1^). FRAP results were calculated and provided as mmol Trolox equivalent per gram (mmol TE g^−1^).

### 2.6. HPLC-DAD-ESI/MS^n^ Analysis of Phenolic Compounds and Methylxanthines

A Hewlett-Packard-1100 HPLC-diode array detector (DAD) chromatograph (Agilent Technologies, Palo Alto, CA, USA), including a quaternary pump, was used to analyze the samples. The mobile phases utilized were 0.1% formic acid in water (solvent A) and 100% acetonitrile (solvent B). The elution gradient employed was isocratic 15% B for 5 min, 15–20% B for 5 min, 20–25% B for 10 min, 25–35% B for 10 min, 35–50% B for 10 min, and column re-equilibration. The chromatographic separation of phytochemicals was conducted at a flow rate of 0.5 mL min^−1^ at 35 °C in a Spherisorb S3 ODS-2 C8 column (Waters, Milford, MA, USA) (3 μm, 150 mm × 4.6 mm i.d.). Based on the different maximum absorbance wavelengths among phytochemicals, the preferred wavelengths for DAD detection were 280 nm (hydroxybenzoic, hydroxycinnamic acids, and caffeine) and 370 nm (flavonols and flavones), even though individual metabolites may have other specific maximum absorbance peaks. The mass spectrometer (MS) was coupled to the HPLC system through the DAD cell output, and the detection was conducted in an API-3200 Qtrap (Applied Biosystems, Darmstadt, Germany) equipped with an ESI source, triple quadrupole-ion trap mass analyzer, and Analyst 5.1 software. Phytochemicals were identified using retention times, UV and mass spectra, fragmentation patterns, and comparison to authentic standards when available. Furthermore, in the case of acyl-quinic acids, the substitution positions were established using the suggested IUPAC numbering system and the previously developed hierarchical keys [22]. For quantitative analysis, quinic derivatives of acyl acids were measured using calibration curves of the respective free acid, and apigenin 6-*C*-glucoside (isovitexin) was used for *C*-glycoside flavones derived from apigenin. Quercetin derivatives were quantified using the quercetin-3-*O*-glucoside curve. Caffeine was quantified using its own standard calibration curves. The concentrations of each phytochemical were given as µg g^−1^ of sample.

### 2.7. Retention Index and Bioaccessibility Calculation

The retention index and bioaccessibility of the phenolic compounds and methylxanthines from the CP, expressed as a percentage, were determined as follows:$$\mathrm{Retention}\;\mathrm{Index}\;\mathrm{or}\;\mathrm{Bioaccessibility}\;(\%)=\frac{C_{\mathrm{Digested}\;\mathrm{fraction}}}{C_{\mathrm{Non}-\mathrm{digested}\;\mathrm{fraction}}}\;\times \;100$$
where C_Digested fraction_ is the concentration of phytochemicals in the soluble digested fraction generated after simulated digestion, whereas C_Non-digested fraction_ is the concentration of phytochemicals in the sample before simulated digestion. Retention index was calculated for oral and gastric stages, whereas bioaccessibility was calculated for intestinal and colonic stages.

### 2.8. Simulated Intestinal Absorption and Bioavailability Calculation

The potential absorption of the phytochemicals found in the CPF and the CPE was estimated in silico. Caco-2 absorption (C2A) and human intestinal absorption (HIA) were computed using canonical SMILES sequences acquired from PubChem (https://palculatedubchem.ncbi.nlm.nih.gov/, accessed on 17 February 2022), using pkCSM-pharmacokinetics (http://biosig.unimelb.edu.au/pkcsm/, accessed on 17 February 2022) and ADMETlab (https://admet.scbdd.com/, accessed on 17 February 2022) cheminformatics free software. The potential bioavailability of phytochemicals was calculated as follows:$$\mathrm{Bioavailability}\;(\%)=\frac{C_{\mathrm{Intestinal}\;\mathrm{fraction}}\;\times \;\mathrm{Absorption}}{C_{\mathrm{Non}-\mathrm{digested}\;\mathrm{fraction}}}\;\times \;100$$
where C_Intestinal fraction_ is the concentration of compounds in the soluble intestinal fraction generated after simulated gastrointestinal digestion, Absorption is the percentage of potentially absorbed phytochemicals estimated in silico for each molecule, and C_Non-digested fraction_ is the concentration of compounds in CPF and CPE before simulated gastrointestinal digestion.

### 2.9. Simulated Colonic Gut Biotransformation

The human gut metabolism of the tentatively identified compounds was predicted in silico using Biotransformer [23]. The metabolism prediction was conducted using the Human Gut Microbial Transformations option. This software predicts small-molecule metabolism by gut microbial enzymes after entering each compound’s canonical SMILES sequences. For the compounds containing unknown glycosides (hexosides), glucosides were considered when selecting the SMILE. Only catabolic reactions (hydrolysis, reduction, dehydroxylation, decarboxylation, oxidation, demethylation, and C-ring fission) were considered, while conjugation reactions (methylation, sulphation, or glucuronidation) were ignored.

### 2.10. Statistical Analysis

Results are expressed as the mean ± standard deviation (SD) of at least three independent experiments (*n* = 3). Data were analyzed by one-way analysis of variance (ANOVA) and post hoc Tukey test for comparisons among digestive phases. *T*-test comparisons were performed between the retention indexes and the bioaccessibility of CPF and CPE. Differences were considered significant at *p* < 0.05.

## 3. Results

### 3.1. The Coffee Pulp Is a Source of Phenolic Compounds and Caffeine

Phenolic compounds and caffeine from the CP were tentatively identified from HPLC-DAD-MS^n^ analysis (Table 1). A total of 17 compounds were identified in the CP, grouped in four distinct phenolic classes, including hydroxybenzoic (3 compounds), hydroxycinnamic derivatives (9 compounds), flavones (1 compound), and flavonols (3 compounds), as well as methylxanthines (caffeine). As hydroxybenzoic acids, gallic (compound **1**), protocatechuic (compound **3**), and vanillic (compound **7**) acids were recognized according to their retention times, UV and MS spectra, and the comparison with genuine standards. Compounds **2**, **4**, **5**, and **6** showed UV spectra like chlorogenic acid. The deprotonated ions of compounds (*m/z* 353) generated a fragmentation pattern including ions at *m/z* 191, associated with quinic acid, at *m/z* 179, linked to [caffeic acid–H]^−^, and at *m/z* 173, coming from [quinic acid-H-H_2_O]^−^. Therefore, compound **2** was tentatively identified as chlorogenic acid (3-*O*-caffeoyl quinic acid), compound **4** as cryptochlorogenic acid (4-*O*-caffeoyl quinic acid, *cis*-isomer), compound **5** as cryptochlorogenic acid (4-*O*-caffeoyl quinic acid, *trans*-isomer), and compound **6** as neochlorogenic acid (5-*O*-caffeoyl quinic acid). Compounds **8** and **13** were identified as caffeic and *p*-coumaric acids by comparing their elution times and UV/MS spectra with available pure standards. Compounds **15** and **17** showed a UV spectrum comparable to caffeic and *p*-coumaric acids, respectively, but they eluted at distinct times. The pseudomolecular ions of compounds **15** and **17** (*m/z* 515 and 337, respectively) produced fragment ions at *m/z* 191, which corresponded to quinic acid, at *m/z* 179, resulting from [caffeic acid–H]^−^, and *m/z* 173 [quinic acid-H_2_O-H]^−^, respectively. Hence, compound **15** was identified as isochlorogenic acid A and compound **17** as 5-*p*-coumaroylquinic acid, as described by Oliva et al. [24]. 

Regarding flavones, compound **10** (*m/z* 593) was identified as vicenin-2 (apigenin 6,8-di-*C*-glucoside) by comparing its features with the standard. Its MS^2^ fragmentations pattern was characteristic of *C*-glycosidic flavones [25]. Quercetin derivatives (compounds **11**, **14** and **16**) presented similar UV spectra (λ_max_ 356 nm) to quercetin hexosides. Compound **11** was identified as quercetin 3,7-dihexoside according to [M-H]^−^ at *m/z* 625. Fragment ion at *m/z* 301, which corresponded to [M−H−(di)hexoside]^−^ or [quercetin−H]^−^. Compound **14** (quercetin 3-*O*-rutinoside) and compound (quercetin 3-*O*-glucoside) were identified by comparing their retention time, mass, and UV-vis chemicals traits compared with commercial pure compounds. Similarly, caffeine (compound **9**) was identified by a λ_max_ at 275 nm through HPLC-DAD-MS^n^ analysis.

The analysis of phenolic compounds in the CPF (Table 2) revealed the presence of hydroxybenzoic acids, standing out protocatechuic acid (20.4%), followed by gallic acid (5.4%) to a lesser extent. Hydroxycinnamic acids were also detected in the CPF, especially from the chlorogenic acid family (cryptochlorogenic (*trans* (9.9%) and *cis* (0.8%)), chlorogenic (0.6%), isochlorogenic acid A (0.6%), neochlorogenic (0.5%), 5-*p*-coumaroylquinic (0.3%), and 5-feruloylquinic (0.2%) acids). Flavonoids identified were flavones (vicenin-2 (0.7%)), and flavonols such as rutin (0.9%), isoquercetin (0.7%), and quercetin 3,7-dihexoside (0.3%). Whereas total phenols represented 41.3%, caffeine accounted for 58.7% of the total compounds noticed in the CPF.

Among the phenolic families found in the CPE (Table 2), hydroxybenzoic acids exhibited the same profile as in the CPF, being protocatechuic acid (22.5%) the main one, followed by gallic acid (5.4%). The concentrations of these phenolic derivatives were 1.7-fold higher in the CPE than in the CPF. Similarly, the fraction related to hydroxycinnamic acids was 1.7-fold higher. Some compounds identified in the CPF, such as neochlorogenic and 5-*p*-coumaroylquinic acids, were not detected in the CPE. However, substantial amounts of chlorogenic acids (cryptochlorogenic (*trans* (10.4%) and *cis* (0.9%)), chlorogenic (0.9%), 5-feruloylquinic (0.6%), and isochlorogenic A (0.5%) acids) were also observed in the CPE. Flavonoids were higher in the CPE than in the CPF (1.4-fold), being identified as vicenin-2 (0.6%), rutin (1.0%), and isoquercetin (0.6%). In contrast to the CPF, quercetin 3,7-dihexoside was not detected in the CPE. Caffeine was the major compound found in the CPE (57.1%) compared to total phenolic compounds, representing 42.9%. Hence, the CPF and the CPE exhibited similar phenolic profiles, even if some compounds remained in the CPF matrix and were not released during the aqueous extraction.

### 3.2. Phenolic Compounds’ Concentration Decreased throughout the Digestion of the Coffee Pulp

After the in vitro simulated digestion, the TPC determined spectrophotometrically increased significantly (*p* < 0.05) in both digested matrices (D-CPF and D-CPE), achieving the maximum TPC in the colonic phase (36.6 mg g^−1^) in case of the D-CPF, and intestinal phase (68.7 mg g^−1^) in the D-CPE (Figure 1A). Subsequently, the TPC in the non-digested CPF (TND-CPF) decreased (*p* < 0.05) throughout the digestion (from 40.7 to 16.8 mg g^−1^). This behavior was also observed in the free and bound phenolic fractions of the non-digested CPF (FND-CPF and BND-CPF). The antioxidant capacity, measured by the ABTS (Figure 1B) and FRAP methods (Figure 1C), showed a similar trend in the D-CPF through both methodologies. The antioxidant capacity increased from the oral to the colonic phase in the D-CPF, increasing at the end of digestion (71.0 and 91.2% for ABTS and FRAP methods, respectively). Conversely, in the TND-CPF, the antioxidant capacity decreased from the oral to the colonic phase, 57.0 and 68.3% for ABTS and FRAP, respectively. This trend was also observed in the FND-CPF, being more accentuated than in the BND-CPF. Nevertheless, the antioxidant capacity in the CPE throughout the in vitro digestion exhibited different behavior by the ABTS (2.9-fold increase) or FRAP (49.5% decrease) methods. Phenolic acids (hydroxybenzoic and hydroxycinnamic acids) are shown in Figure 1D.

Concerning the CPF, phenolic acids decreased 19.9% (*p* < 0.05) at the end of digestion compared to the non-digested flour, although they exhibited a noteworthy increase (1.4-fold) from the oral to the intestinal phase. Contrariwise, phenolic acids remained stable in the CPE, except during the intestinal phase since their content was significantly reduced (40.0%) compared to the non-digested CPE and the other digestive phases. In general, hydroxybenzoic and hydroxycinnamic acids showed the same behavior individually in the CPF and the CPE. The concentration of flavonoids (flavones and flavonols) in the CPF decreased (32.2%) after colonic digestion (Figure 1E). Altogether, flavonoids from the CPF released during the oral phase (35.8%) increased during gastric (59.4%) and intestinal (54.0%) phases, followed by a decrease in the colonic phase (17.7%). In contrast, flavonoids in the CPE were ultimately released during the oral (100%) and gastric (93.0%) phases, but they were not detected during the intestinal and colonic phases. Total phenolic compounds from the CPF were significantly liberated during the oral phase and increased in the intestinal phase (1.4-fold, *p* < 0.05), but a drastic decrease was detected (37.1%) from the intestinal to the colonic phase (Figure 1F). In contrast, total phenolic compounds from the CPE were fully released during the oral phase. The digestive process resulted in a drastic reduction in the concentration of total phenolic compounds, especially during the intestinal phase (45.8%), although they slightly increased from the intestinal to colonic phase (1.5-fold). Caffeine release through the in vitro digestion differed depending on the matrix type (Figure 1F). Then, in the CPF, the caffeine concentration increased from the oral to the colonic phase (44.4%). However, in the CPE, caffeine did not undergo release nor degradation (*p* > 0.05) during the simulated digestion. Interestingly, the total phenolic compounds and caffeine trends were similar in each matrix type (flour and extract). A decreasing proportion of phenolic compounds in the CPF was shown throughout the digestion, while caffeine proportion increased (Figure 1G). Hydroxybenzoic and hydroxycinnamic acid derivatives maintained their percentage until the colonic phase, in which flavanols and flavones were drastically decreased. In the CPE, the phenolic distribution was similar to that of the CPF, but in the colonic phase, the proportion of hydroxycinnamic acids increased, lowering the percentage of caffeine since flavanols and flavones were detected (Figure 1H). Therefore, the potential absorption of these phenolic compounds revealed that hydroxybenzoic acids were stable throughout the digestion showing a good absorption, while hydroxycinnamic acid derivatives displayed a poor absorption (C2A and HIA). In contrast, the proportion of caffeine increased over the simulated digestion and absorbed phases, excluding the colonic phase.

### 3.3. Phenolic Acids and Caffeine Were Highly Bioaccessible, Whereas Flavonoids Were Degraded

Total phenolics and caffeine released from the CPF showed lower concentrations during the oral phase than in the non-digested sample (Table 2). Hydroxybenzoic acids (gallic and protocatechuic acid) presented a retention index of 60.1 and 61.4%, respectively (Table 3). In the case of hydroxycinnamic acids, the retention rate ranged from 6.7 to 62.6%. Vicenin-2, the only flavone noticed in the CPF, achieved a retention index of 27.8%, while flavonols exhibited retention indexes between 30.7 and 45.4%. Finally, caffeine, the most abundant compound in the CPF (4730.6 µg g^−1^), presented a retention rate of 42.6%. Regarding the CPE, phenolic compounds and caffeine were completely liberated during the oral phase; their concentration exhibited no significant (*p* < 0.05) differences between the oral phase and the non-digested material (Table 2). In the gastric phase, the concentration of hydroxybenzoic acids from CPF did not show significant differences (*p* < 0.05) in comparison to the oral phase, exhibiting high gastric retention indexes, being 53.2% for gallic acid and 74.9% for protocatechuic acid (Table 3). Concerning hydroxycinnamic acid derivatives, the concentration of several compounds increased significantly, such as chlorogenic acid (1.3-fold), cryptochlorogenic acid (*cis*) (1.4-fold), cryptochlorogenic acid (*trans*) (1.4-fold), isochlorogenic A acid (1.6-fold), and 5-*p*-couramaroylquinic acid (2.9-fold) as shown in Table 2, with gastric retention indexes ranging from 19.7 to 88.4%. In contrast to the oral phase, 5-ferulolyquinic acid was found in the CPF gastric phase.

Regarding the CPE, gallic acid decreased significantly (*p* < 0.05) from the oral to gastric phase by 38.6%, while protocatechuic acid remained stable. Hydroxycinnamic acids such as cryptochlorogenic acid (*trans*) decreased significantly (24.6%). No significant modifications were observed in the flavonoid fraction, except for isoquercetin (13.6% decrease). As observed in the oral phase, the gastric retention index of the compounds mentioned above was higher in the CPE than in the CPF. In the intestinal phase, the concentration of protocatechuic acid in the CPF increased significantly from the gastric to the intestinal phase (1.2-fold), representing a bioaccessibility of 91.9% (Table 3). Concerning hydroxycinnamic acids, except for cryptochlorogenic acid (*trans*), their concentrations significantly increased in the intestinal phase compared to the gastric phase. Moreover, all hydroxycinnamic acids exhibited high bioaccessibility (over 71.2%), particularly chlorogenic acid (123.2%) and cryptochlorogenic acid (*cis*) (122.7%). Quercetin 3,7-dihexoside, rutin, and isoquercetin did not differ from the gastric phase, achieving a bioaccessibility of 62.8, 58.4, and 57.3%, respectively. Once again, a remarkable caffeine bioaccessibility was noted (82.2%). Hydroxybenzoic acids in the CPE revealed a high bioaccessibility, although they underwent degradation during the intestinal phase (52.2 and 27.1% for gallic and protocatechuic acids, respectively). Some hydroxycinnamic acids also suffered drastic degradation from the gastric to the intestinal phase, such as cryptochlorogenic (*trans*) (42.7%). 5-feruloylquinic acid was not even detected in the intestinal phase. Flavonoids were also not detected in the intestinal phase. Chlorogenic, cryptochlorogenic (*cis*), and isochlorogenic A acids showed a high bioaccessibility (86.5, 113.8, and 97.7%, respectively), along with caffeine (84.9%). In general, the intestinal bioaccessibility of phenolic compounds seemed to be higher in the CPF than in the CPE (Appendix A). During the colonic stage, gallic and protocatechuic acid concentrations decreased by 27.7 and 32.6%, respectively (Table 2), compared to the intestinal phase, so their bioaccessibility was lower (22.1 and 62.0%, respectively). The concentration of cryptochlorogenic (*trans*) isochlorogenic A, and 5-*p*-coumaroylquinic acids decreased considerably (*p* < 0.05) during this phase. Then, these hydroxycinnamic acids exhibited low bioaccessibility (14.7–41.2%). Flavones and flavanols decreased from the intestinal to the colonic phase. Quercetin 3,7-dihexoside and rutin were not detected during the colonic phase. A reduction in the caffeine content was also observed after the colonic stage but still exhibited a high bioaccessibility (76.5%). In turn, the analysis of the CPF after colonic digestion revealed the presence of 3 new compounds, vanillic, caffeic, and *p*-coumaric acids, which were not detected in the non-digested samples. Concerning the CPE in the colonic phase, gallic acid increased significantly with respect to the intestinal phase, showing a high bioaccessibility (97.4%). The concentration of protocatechuic acid was not affected, so this acid also exhibited a high bioaccessibility (74.7%). In the case of hydroxycinnamic acids, they were either found in traces or not detected. Caffeine preserved a high bioaccessibility during this stage (83.0%). In summary, the bioaccessibility of phenolic compounds in the colonic phase was higher in the CPF than in the CPE, while caffeine remained bioaccessible in both.

### 3.4. Caffeine and Protocatechuic Acid Were the Main Compounds Absorbed in the Intestine after Gastrointestinal Digestion

After the intestinal phase of the CPF and the CPE digestion, the potential bioavailability of caffeine and phenolic compounds was studied through two in silico models (C2A and HIA) (Table 2 and Table 3). Phenolic compounds exhibited low absorption in comparison to caffeine. The bioavailability values obtained using the C2A model were higher than those obtained using the HIA model for hydroxybenzoic acids (1.2- to 1.4-fold), caffeine (1.1-fold), and some flavonoids (1.4- to 1.5-fold). In contrast, the HIA model provided higher bioavailability values for hydroxycinnamic acids than the C2A model (1.6 to 3.4-fold). Concerning the CPF, the most bioavailable phenolic compounds were the hydroxybenzoic acids, highlighting protocatechuic acid (58.0–80.3%), followed by hydroxycinnamic acid derivatives, especially chlorogenic acid, which could reach 42.4% of bioavailability. The flavonoid fraction presented low bioavailability (less than 25.3 %). The results showed that phenolic compounds from the CPF were more bioavailable than those from the CPE. During the intestinal phase of the CPE digestion, some phenolic acids such as 5-feruloylquinic acid, vicenin-2, rutin, and isoquercetin were not detected, while the protocatechuic acid stood out for its high bioavailability (44.6–61.7%). From the results, hydroxybenzoic acids appeared to be more bioavailable than hydroxycinnamic acids, and these latter, in turn, were more bioavailable than flavonoids (Appendix A). In contrast to phenolic compounds, caffeine was highlighted for its high bioavailability after intestinal digestion in both matrices (72.2–83.1%).

### 3.5. The Matrix of the Coffee Pulp Influenced the Behavior of Phenolic Compounds during Digestion

A principal components analysis and a hierarchical cluster analysis coupled with a heatmap provided an integrated vision of the behavior of the CPF and the CPE during digestion (Figure 2). We found 13 different components explaining the variability among samples. The two principal components could explain 72.4% of the whole variability (Figure 2A). The first component (49.0% of the variability) mainly comprised the effects of cryptochlorogenic (*trans*), 5-feruloylquinic, and protocatechuic acids, rutin, FRAP, and partially caffeine (contributes to both components one and two). The second component (23.4% of the variability) included the TPC and partially caffeine effects. Phenolic compounds and caffeine in the CPF were sequentially released from the matrix from the oral to the intestinal phase and diminished in the colonic phase. Furthermore, the antioxidant capacity of CPF digested fractions increased over digestion. Conversely, phenolic compounds in the CPE were fully released in the oral phase, were stable to the acidic condition of the gastric phase, and then their concentration was reduced in the intestinal and colonic phases. However, the poor absorption of phenolic compounds from both matrices yielded similar bioavailable samples. Then, we can observe that, independently of the initial composition of CPF and CPE, their bioavailable fractions were similar, all of them grouped in the left panels.

Complementarily, the hierarchical cluster analysis depicted the grouping of the samples according to their composition (Figure 2B). The first group included non-digested CPE and the oral and gastric fractions. In the second group, we found CPF samples from the non-digested to the intestinal phase, and the third group clustered the colonic and the bioavailable phases of CPF and CPE. Altogether, multivariate analysis demonstrated that gastrointestinal digestion modified the composition of the CP, following a matrix-dependent behavior, bridging the initial gap between the composition of CPF and CPE.

### 3.6. Non-Absorbed Phenolic Compounds Might Undergo Colonic Biotransformation Yielding Small and Potentially More Adsorbable Phenolic Metabolites

Phenolic compound colonic metabolism by the gut microbiota was investigated *in silico,* and the main pathways involved in the biotransformation of the CP’s phenolics were summarized in Figure 3. Chlorogenic acids are mainly hydrolyzed into caffeic acid (CA), which subsequently can be reduced, resulting in dihydrocaffeic acid (dhCA), or dehydroxylated into *p*-coumaric (*p*-CouA) and *m*-coumaric (*m*-CouA) acids. Protocatechuic acid (PCA), although primarily absorbed in the small intestine, could also appear in the colon, derived from the oxidation of dihydrocaffeic and 3,4-hydroxyphenylacetic (3,4-HPAA) acids, the dihydroxylation of gallic acid (GA), or the demethylation of vanillic acid (VA). Then, PCA can be dehydroxylated, yielding hydrocinnamic (hCiA), *p*-salicylic (*p*-SA), and *m*-salicylic acids (*m*-SA). The smallest metabolite produced from the colonic metabolism of phenolic acids might be benzoic acid (BA).

Quercetin glycosides such as quercetin 3,7-diglucoside (Q3,7G), quercetin 3-glucoside (Q3G), quercetin 7-glucoside (Q7G), and rutin (RUT), were hydrolyzed into quercetin (Q), which in turn was dehydroxylated into 3,3′,5,7-tetrahydroxyflavone (THFL) and kaempferol (KMP) and reduced into dihydroquercetin (dhQ). On the other hand, vicenin-2 (VIC2) was hydrolyzed into apigenin 6-*C*-glucoside (API6G), which was subsequently hydrolyzed into apigenin (API). Apigenin was transformed into naringenin (NAR) and chrysin (CHRY) by reduction and dehydroxylation, respectively. After further transformations (reduction, dehydroxylation, and C-ring fission), flavonols and flavones yielded different chalcones, which hydrolysis resulted in hydroxyphenylpropionic, hydroxyphenylpyruvic, and hydroxyphenylacetic acids. Phenylacetic (PAA) and hydrocinnamic acids were the primary metabolites obtained from the colonic metabolism of flavonols and flavones, respectively, ultimately generating benzoic acid.

Colonic biotransformation could influence the absorption of the CP’s phenolic compounds due to the generation of different compounds from those found in the intestinal phase of the simulated digestion. Distinct pathways conduced to the formation of small and low molecular weight metabolites with a higher potential absorption in both C2A and HIA models (Appendix A). Those low molecular weight phenolic compounds (i.e., phenylacetic or benzoic acid) might be fully absorbed in the intestine in contrast to the parent molecules (Appendix A). Quercetin and apigenin glycosides exhibited a potential intestinal absorption below 32%. Caffeoylquinic acids could be absorbed in around 26.3–34.2%. In contrast, protocatechuic acid might be better absorbed (63.1%). Therefore, non-absorbed phenolic compounds might undergo colonic biotransformation yielding small and potentially more adsorbable phenolic metabolites. Colonic metabolism might be critical for the absorption of the phenolic metabolites derived from the CP.

## 4. Discussion

Sustainability awareness is increasing, and the valorization of food by-products, including the CP, can represent a suitable strategy to promote an eco-friendlier coffee industry and the production of high-value-added food ingredients. This approach can promote a more productive and sustainable food system. The CP is rich in dietary fiber and phytochemicals, mainly phenolic compounds and caffeine, with health-promoting properties that can contribute to preventing some diseases [3]. In a previous report, we validated the safe intake of the CP. Despite the high concentration of caffeine in the CPF and the CPE, no signs of toxicity were observed after their acute (2 g kg^−1^ day^−1^, 1 day) and chronic (1 g kg^−1^ day^−1^, 90 days) administration in mice [26]. In the present study, we evaluated for the first time the impact of simulated digestion on the bioaccessibility and bioavailability of phenolic compounds and caffeine from the CPF and the CPE. Like other coffee by-products, the CP displayed a high concentration of caffeine and chlorogenic, protocatechuic, and gallic acids. A significant fraction of flavonoids was also identified, mainly composed of quercetin derivatives. Caffeine was the main phytochemical found in the CP, standing out among all phenolic compounds, comparable to the sum of all of them. Other coffee by-products studied previously as coffee parchment were composed mainly of chlorogenic, vanillic, protocatechuic, and *p*-coumaric acids [6]. Similarly, chlorogenic, protocatechuic, and gallic acids were the main phenolics identified in the coffee husk [27], whereas in the coffee silverskin, chlorogenic acid and caffeine were the main compounds identified [28]. Caffeine and chlorogenic acids are two of the most important bioactive compounds regardless of the coffee by-product. The concentration of phenolic compounds and caffeine was higher in the CPE than in the CPF since the freeze-dried process conducted in the aqueous extract concentrated those components. The extraction of phenolic compounds is related to their solubility degree and chemical structure [29]. Hydroxybenzoic acid derivatives exhibit high solubility due to the presence of hydroxyl groups in their structure, facilitating water interactions and enhancing extraction yields [30]. The differences in the phenolic profile of the CPF and the CPE were mainly due to the matrix type. Whereas in the CPF, 13 phenolic compounds and caffeine were found, in the CPE, just 10 phenolic compounds and caffeine were identified, indicating the differential migration of phytochemicals from the CPF. Some of the phenolics (neochlorogenic acid, 5-*p*-coumaroylquinic acid, and quercetin 3,7-dihexoside) could not be released or were degraded during the heat-assisted extraction [7]. Although heat primarily promoted the extraction of phenolic compounds from the CP matrix to the aqueous media, existing evidence also supports that some compounds may be transformed or degraded when subjected to high temperatures [31]. Phenolic compounds can be linked to cell wall components providing specific benefits associated with dietary fiber-linked compounds [2]. Phenolic compounds are sometimes covalently linked to the cell wall polysaccharides by ester bonds, especially hydroxycinnamic acids. Consistently, vanillic, *p*-coumaric, and caffeic acids were not detected until the colonic phase, as they were probably released from the fiber matrix by cell wall-degrading enzymes [32].

During the in vitro gastrointestinal digestion of the CPF and the CPE, the TPC and the ABTS antioxidant capacity measured spectrophotometrically increased throughout digested fractions, accompanied by a drastic decrease in the corresponding non-digested flour fractions (ND-CPF). The release of phenolic acids from the non-digested insoluble fiber fraction may explain the higher values obtained by Folin–Ciocalteu and ABTS methods. The Folin–Ciocalteu reagent interacts with other compounds released during the simulated gastrointestinal digestion, potentially overestimating the results, and does not correlate well with the sum of individual phenolic compounds evaluated chromatographically. Nevertheless, since phenolic compounds present the highest antioxidant capacity, these methods can estimate the antioxidant capacity [33]. Gastrointestinal digestion could increase the antioxidant capacity of phenolic compounds by changing their molecular weight and chemical structure under simulated conditions [33,34].

The oral retention indexes indicated that phenolic compounds released from CPF were mainly free phenolics. Hence, CPE phenolic compounds were fully released during the oral phase. During gastric digestion, the release of phenolic compounds from the CPF is possibly associated with the hydrolysis of some phenolic compounds bound to other food constituents, influenced by digestion time and acid pH [35]. As digestion advances, the particle size of the food is reduced, facilitating the release of phenolic compounds. Most of these compounds are liberated during gastric digestion, where they might be absorbed in their free form [36]. No additional release of phenolic compounds occurred during gastric digestion of the CPE, demonstrating that all phenolic compounds had been solubilized during the oral phase. Phenolic compounds’ release from the CPF increased during the intestinal phase. The pancreatic enzymes and the neutral pH, which may promote phenolics’ deprotonation, may also have facilitated the release of phenolic compounds bound to the food matrix [37]. Besides, the increase in some phenolic compounds could be a consequence of the transformation of compounds such as protocatechuic acid, which increased its content probably due to the dehydroxylation of gallic acid [38]. Conversely, losses in the phenolic content of the CPE after intestinal digestion may be associated with the degradation and transformation reactions occurring under intestinal conditions [39].

The beneficial properties of phenolic compounds depend on their bioaccessibility and bioavailability. Although some phenolic compounds, especially hydroxycinnamic acids, exhibited high bioaccessibility in both matrices, low bioavailability was estimated using in silico models for most of the phenolic compounds studied. About 5–10% of consumed phenolic compounds are expected to be absorbed in the small intestine. Although further studies about the absorption mechanisms of phenolic compounds are required, it is assumed that low molecular weight phenolic compounds can be absorbed by passive transcellular diffusion, while phenolic compounds with high molecular weight are less absorbed in the small intestine. These components were mainly found as polar glycosides, thus limiting their absorption. However, hydrolyzed phenols can be converted into aglycones and reach the liver through the portal vein [36]. Knowledge about the bioaccessibility of hydroxybenzoic acids is limited. Although gallic acid showed low in vitro bioaccessibility and in silico bioavailability, several human studies have shown that this acid is quickly absorbed, exhibiting excellent bioavailability [40]. Protocatechuic acid showed the highest bioavailability among all the compounds studied and could be absorbed through the intestinal epithelium reaching the systemic circulation. Nevertheless, protocatechuic acid may undergo structural modifications such as glucuronidation and sulfation before absorption, compromising its intestinal absorption and bioactivity [41].

Hydroxycinnamic acids exhibited great bioaccessibility, being able to be absorbed in the stomach and intestine by different mechanisms. The exact pathway of hydroxycinnamic acids absorption is not precisely known yet. Thus various studies suggest that these acids are taken up by a combination of passive transmembrane diffusion and monocarboxylic acid transporter systems, which can convey phenolic compounds containing a monoanionic carboxylic acid group and a nonpolar side chain or aromatic hydrophobic unit in their structure [42]. Their low bioavailability is attributed to the fact that hydroxycinnamic acids are not usually found free since they are generally esterified with quinic acids, tartaric acids, or sugar derivatives, delaying their absorption [43]. Although hydroxycinnamic acids can be absorbed in the small intestine after being hydrolyzed by mucosal esterases, they are de-esterified mainly by the colonic microbiota [44]. Grzelczyk et al. demonstrated that hydroxycinnamic acids from coffee are poorly released in the stomach, and they can reach higher levels in the large intestine due to the action of the microbiota [45]. Flavonoids presented low bioaccessibility and bioavailability, especially in the CPE, where they were not detected in the intestinal fraction. Most flavonoids are present in their glycoside form in the food matrix; therefore, the sugar moiety must be removed for flavonoid absorption. The released aglycone can passively diffuse into the epithelial cells [46]. However, the bioavailability of flavonoids may be influenced by their structure and interactions, as they interact with digestive enzymes and food components in the diet. Interaction with protein may reduce the bioavailability of flavonoids, while the opposite was noted when flavonoids are co-administered with starches and fats [47]. Therefore, the CPE soluble fiber may have limited the solubilization of phenolic compounds during digestion [32]. In general, phenolic compounds from the CPF displayed moderate bioaccessibility, as they probably were stabilized by being bound to the matrix until the intestinal phase, where the enzymatic activity liberated them. However, phenolic compounds from the CPE presented less bioaccessibility. Despite being released during the oral phase, they probably suffered degradation or transformation throughout digestion. The bioaccessibility and stability of caffeine and phenolic compounds in the CP seemed to depend on their chemical class, being the highest in hydroxycinnamic acid derivatives and caffeine. Phenolic compounds from both matrices showed low bioavailability, unlike caffeine, a stable molecule with hydrophilic and sufficiently lipophilic properties, conferring it the capacity to permeate through biological membranes, resulting in high bioaccessibility and bioavailability. This methylxanthine is rapidly absorbed from the gastrointestinal tract and distributed throughout the body [48]. Compared to flavonoids, the high bioaccessibility of phenolic acids may be attributed to their high polarity and instability under gastrointestinal digestion conditions [49]. The structural complexity and polymerization of phenolic compounds justify their low absorption in the small intestine, hindering their action as therapeutic agents.

Despite some compounds’ low bioaccessibility and bioavailability, non-absorbed phenolic compounds can reach the large intestine, where they are metabolized by enzymes of the intestinal microbiota for their conversion into more bioavailable molecules of lower molecular weight. The main reactions involved in the biotransformation of phenolic compounds are hydroxylation, oxidation, decarboxylation, methylation, isomerization, hydration, dehydrogenation, and glycosylation [15]. Generally, the lower concentration of phenolic compounds released throughout the colonic phase is due to their degradation or transformation into other compounds. Caffeic and *p*-coumaric acids not identified previously were probably generated by the degradation of chlorogenic acids [43]. Among hydroxybenzoic acids, protocatechuic acid presented high bioaccessibility and bioavailability. Some studies have indicated that protocatechuic acid content in the colonic phase could be higher than the amount ingested since the colonic microbiota can produce highly bioavailable protocatechuic acid from flavonoids such as anthocyanins, procyanidins, flavanols, and other phenolic acids [41].

Hydroxycinnamic acids not absorbed in the intestine are metabolized in the colon mainly by lactobacilli and bifidobacteria, which break the ester bonds linking hydroxycinnamic acids to other molecules through their esterases [44]. Furthermore, hydrolysis of hydroxycinnamic acids can be enhanced by the action of other enzymes such as xylanases which digest cell wall hemicelluloses. Consequently, esterases can better access hydroxycinnamic acids, de-esterifying hydroxycinnamates into ferulic, sinapic, and *p*-coumaric acids, among other phenolic compounds [44]. According to our in silico study, chlorogenic acids were converted into 3,3- and 3,4-hydroxyphenylpropionic acids. The gut microbiota is highly responsible for metabolizing chlorogenic acids into caffeic acid and transforming it into hydroxyphenylpropionic acids. Benzoic acid might be the final metabolite proceeding from the metabolism of 3-hydroxyphenylpropionic through its dehydroxylation into 3-phenylpropionic acid and the β-oxidation of 3-phenylpropionic acid [42]. 

Equivalently, flavonoids not absorbed in the small intestine can also reach the colon bound to a glycoside hydrolyzed by glycosidases secreted by the colonic microbiota [36]. Rutin, which represents one of the most consumed flavonoids, can be transformed by the gut microbiota into quercetin-3-glucoside and quercetin, which is then metabolized into 3,4-dihydroxyphenylacetic, protocatechuic, and *p*-salicylic acid acids. Correspondingly, apigenin is mainly transformed into 3-(4-hydroxyphenyl)propionic acid [50]. These results are consistent with our in silico study since all the quercetin, and apigenin glycosides were transformed into protocatechuic, phenylacetic, and 3-(4-hydroxyphenyl)propionic acids, which were subsequently metabolized into hydrocinnamic, *p*- and *m*-salicylic, and benzoic acids. As observed, low molecular weight phenolic metabolites of the colonic microbiota are the ones that can be absorbed and reach the bloodstream and peripheral tissues eliciting bioactive effects [51]. Notwithstanding, further investigations are required to study the bioavailability of these compounds from the CP and the bioactivities of the generated metabolites due to the limitations of in silico model.

## 5. Conclusions

This research brings about a new understanding of the gastrointestinal behavior of the CP phenolic compounds and caffeine under simulated conditions. The bioaccessibility, potential bioavailability, and colonic biotransformation of the phytochemicals from the CP have been investigated for the first time. Phenolic compounds decreased throughout the digestive process, while caffeine levels remained stable. The matrix type of the CP has a remarkable role in the release of phytochemicals; phenolic compounds from the CPF were more bioaccessible and bioavailable than those from the CPE, mainly due to their interactions with the food matrix. Despite the high quantities of caffeoyl quinic acids in coffee and its by-products, results indicated that protocatechuic acid, along with caffeine, might be the most important metabolite absorbed following CP consumption. The new knowledge generated on the non-bioaccessible phenolic compounds through the in silico prediction of their colonic metabolism revealed that non-absorbed phenolic compounds might be metabolized and transformed by the intestinal microbiota into other lower molecular weight compounds. These metabolites that reach the colon may exert beneficial effects on the intestinal microbiota, play a significant role in intestinal health, and be further absorbed and distributed in the organism. According to the results, the CP could be used as an antioxidant food ingredient since it is a source of bioaccessible and potentially bioavailable phytochemicals, which might exert potential health-promoting properties in the organism.

## Figures and Tables

**Figure 1 antioxidants-11-01818-f001:**
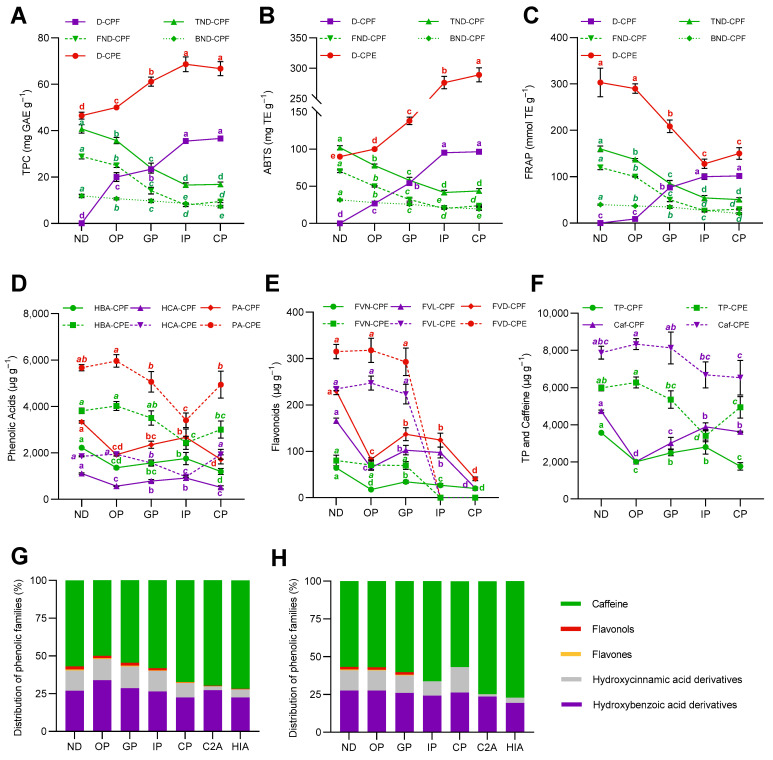
Effect of in vitro digestion of coffee pulp flour (CPF) and coffee pulp extract (CPE) on the total phenolic content (**A**), ABTS (**B**), and FRAP (**C**) antioxidant capacity from the digested (D) and non-digested (ND) coffee pulp, including free (FND), bound (BND), and total (TND) phenolic fractions. Behavior of total phenolic acids (PA) (hydroxybenzoic acids (HBA) + hydroxycinnamic acids (HCA)) (**D**), flavonoids (FVD) (flavones (FVN) + flavonols (FVL)) (**E**), and total phenolics (TP) and caffeine (**F**) throughout the simulated gastrointestinal digestion phases. Distribution of the different phenolic families (flavonols, flavones, hydroxybenzoic acids, and hydroxycinnamic acids) and caffeine in the CPF (**G**) and the CPE (**H**) throughout simulated gastrointestinal digestion. The results are reported as mean ± SD (*n* = 3). Points with different letters significantly (*p* < 0.05) differ according to ANOVA and Tukey’s multiple range test.

**Figure 2 antioxidants-11-01818-f002:**
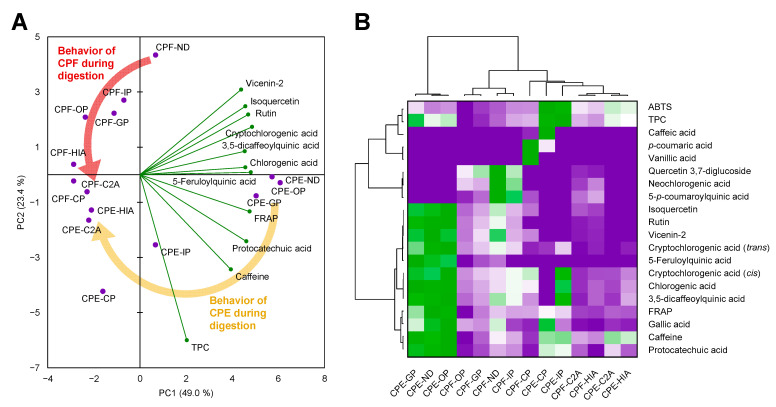
Biplot (scores of samples and load factors of each variable) of the principal component analysis (PCA) (**A**) and agglomerative hierarchical cluster analysis coupled to heatmap (from the lowest (

) to the highest (

) value for each parameter) (**B**) illustrating the behavior of phenolic compounds and caffeine from the coffee pulp during simulated gastrointestinal digestion.

**Figure 3 antioxidants-11-01818-f003:**
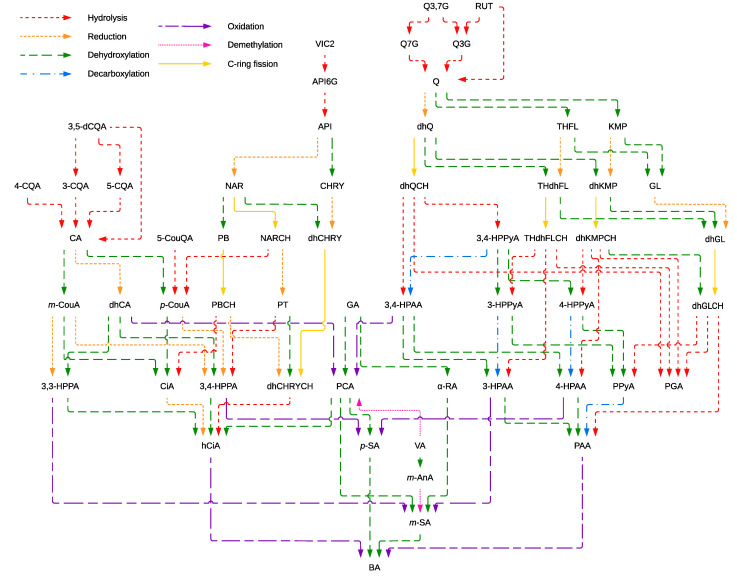
Proposed pathways involved in the colonic metabolism of phenolic compounds from the coffee pulp predicted in silico. The full names of phenolic compounds are defined in Appendix A.

**Table 1 antioxidants-11-01818-t001:** Identification of phenolic compounds and methylxanthines in the coffee pulp flour (CPF) and extract (CPE) by HPLC-DAD-MS *.

Comp.	*R_t_* (min)	*λ*_max_ (nm)	Molecular Ion [M-H]^−^ (*m/z*)	Fragments MS^2^	Tentative Identification	Common Name
**1**	4.38	270	169	–	3,4,5-trihydroxybenzoic acid	Gallic acid
**2**	5.17	326	353	191(100), 135(83), 179(54), 173(5), 161(7)	3-*O*-caffeoyl quinic acid	Chlorogenic acid
**3**	6.08	260, 294	153	109(70)	3,4-dihydroxybenzoic acid	Protocatechuic acid
**4**	7.28	326	353	191(8), 179(4), 173(3), 161(12), 135(94)	4-*O*-caffeoyl quinic acid *cis* isomer	Cryptochlorogenic acid (*cis*)
**5**	8.12	326	353	191(100), 179(18), 173(6), 161(21), 135(12)	4-*O*-caffeoyl quinic acid *trans* isomer	Cryptochlorogenic acid (*trans*)
**6**	10.05	328	353	191(39), 179(9), 173(49), 155(33), 135(38)	5-*O*-caffeoyl quinic acid	Neochlorogenic acid
**7**	11.35	252, 290	167	–	4-hydroxy-3-methoxybenzoic acid	Vanillic acid
**8**	11.70	323	179	–	3,4-dihydroxycinnamic acid	Caffeic acid
**9**	12.99	275	–	–	1,3,7-trimethylxanthine	Caffeine
**10**	12.30	340	593	353(100), 383(70), 297(55), 473(50), 503(15)	Apigenin 6,8-di-*C*-glucoside	Vicenin-2
**11**	16.36	355	625	301(100)	Quercetin 3,7-dihexoside	–
**12**	16.60	328	367	193(6), 191(100), 173(4), 134(9)	5-feruloylquinic acid	–
**13**	17.40	233, 314	163	119(70)	4-hydroxycinnamic acid	*p*-coumaric acid
**14**	19.80	356	609	301(100)	Quercetin-3-*O*-rutinoside	Rutin
**15**	21.77	328	515	353(100), 335(5), 191(97), 179(79), 173(9), 161(5), 135(37)	3,5-dicaffeoylquinic acid	Isochlorogenic acid A
**16**	21.81	356	463	301(100)	Quercetin-3-*O*-glucoside	Isoquercetin
**17**	23.49	314	337	191(100), 173(10), 163(69)	5-*p*-coumaroylquinic acid	–

* The values in brackets indicate the relative intensity of fragments.

**Table 2 antioxidants-11-01818-t002:** Evolution of the concentration of individual phenolic compounds and methylxanthines (μg g^−1^) in the non-digested and digested coffee pulp flour (CPF) and extract (CPE) and its potential Caco-2 and human intestinal absorption throughout the phases of the gastrointestinal digestion.

Compounds	ND	OP	GP	IP	CP	C2A	HIA
* **Coffee pulp flour** *							
*Hydroxybenzoic acid derivatives*							
Gallic acid	469.4 ± 20.2 ^a^	281.9 ± 24.0 ^b^	249.7 ± 11.7 ^b^	143.8 ± 20.1 ^c^	103.9 ± 13.0 ^d^	77.5 ± 10.8 ^de^	62.2 ± 8.7 ^e^
Protocatechuic acid	1757.5 ± 7.3 ^a^	1079.2 ± 19.6 ^d^	1315.6 ± 81.9 ^bcd^	1615.8 ± 239.5 ^ab^	1089.2 ± 117.1 ^cd^	1411.4 ± 209.3 ^abc^	1020.1 ± 151.2 ^d^
Vanillic acid	nd	nd	nd	nd	7.6 ± 0.3 ^a^	–	–
*Hydroxycinnamic acid derivatives*							
Chlorogenic acid	52.8 ± 6.3 ^b^	29.8 ± 1.9 ^d^	40.2 ± 5.1 ^c^	65.0 ± 3.9 ^a^	57.5 ± 5.1 ^ab^	9.9 ± 0.6 ^e^	22.4 ± 1.4 ^d^
Cryptochlorogenic acid (*cis*)	66.2 ± 5.0 ^bc^	41.4 ± 2.0 ^d^	58.5 ± 5.1 ^c^	81.2 ± 12.4 ^ab^	94.0 ± 12.7 ^a^	12.3 ± 1.9 ^e^	21.3 ± 3.3 ^e^
Cryptochlorogenic acid (*trans*)	853.2 ± 14.3 ^a^	459.1 ± 5.9 ^c^	635.1 ± 42.4 ^b^	676.7 ± 73.4 ^b^	180.3 ± 20.9 ^d^	69.7 ± 7.6 ^d^	177.7 ± 19.3 ^d^
Neochlorogenic acid	38.9 ± 2.7 ^a^	15.4 ± 1.0 ^c^	12.8 ± 1.7 ^cd^	27.7 ± 3.6 ^b^	nd	4.2 ± 0.5 ^e^	9.5 ± 1.2 ^d^
Caffeic acid	nd	nd	nd	nd	110.0 ± 15.3 ^a^	–	–
5-feruloylquinic acid	20.1 ± 1.3 ^a^	nd	13.8 ± 0.5 ^b^	nd	nd	–	–
*p*-coumaric acid	nd	nd	nd	nd	50.6 ± 7.8 ^a^	–	–
Isochlorogenic acid A	49.0 ± 1.1 ^a^	14.2 ± 1.5 ^d^	22.1 ± 2.0 ^c^	38.6 ± 4.2 ^b^	20.2 ± 2.6 ^c^	3.9 ± 0.4 ^e^	13.2 ± 1.4 ^d^
5-*p*-coumaroylquinic acid	28.5 ± 0.0 ^a^	1.9 ± 0.7 ^e^	5.6 ± 0.7 ^cd^	23.3 ± 3.4 ^b^	4.2 ± 0.4 ^de^	5.7 ± 0.8 ^cd^	8.9 ± 1.3 ^c^
*Flavones*							
Vicenin-2	64.3 ± 2.0 ^a^	17.9 ± 0.4 ^d^	34.3 ± 3.5 ^b^	26.9 ± 3.5 ^c^	20.1 ± 1.6 ^d^	3.2 ± 0.4 ^e^	5.0 ± 0.7 ^e^
*Flavonols*							
Quercetin 3,7-dihexoside	25.9 ± 1.1 ^a^	11.8 ± 0.8 ^c^	17.1 ± 0.1 ^b^	16.3 ± 1.3 ^b^	nd	1.8 ± 0.1 ^d^	1.2 ± 0.1 ^d^
Rutin	79.3 ± 2.1 ^a^	34.1 ± 1.1 ^d^	54.8 ± 6.0 ^b^	46.3 ± 4.8 ^c^	nd	6.3 ± 0.7 ^e^	10.3 ± 1.1 ^e^
Isoquercetin	60.9 ± 2.6 ^a^	18.7 ± 1.4 ^c^	30.8 ± 4.0 ^b^	34.9 ± 5.3 ^b^	20.7 ± 2.4 ^c^	15.4 ± 2.3 ^cd^	11.1 ± 1.7 ^d^
*Methylxanthines*							
Caffeine	4730.6 ± 60.9 ^a^	2013.2 ± 0.1 ^e^	3009.3 ± 314.8 ^d^	3887.1 ± 220.6 ^b^	3618.9 ± 29.8 ^bc^	3805.5 ± 215.9 ^bc^	3462.8 ± 196.5 ^c^
* **Coffee pulp extract** *							
*Hydroxybenzoic acid derivatives*							
Gallic acid	684.5 ± 40.4 ^a^	759.5 ± 78.1 ^a^	466.0 ± 28.5 ^b^	222.9 ± 23.3 ^c^	666.4 ± 74.0 ^a^	120.1 ± 12.5 ^cd^	96.4 ± 10.1 ^d^
Protocatechuic acid	3132.1 ± 75.2 ^a^	3266.6 ± 111.8 ^a^	3037.3 ± 281.7 ^a^	2212.7 ± 258.2 ^b^	2338.4 ± 294.1 ^b^	1932.9 ± 225.5 ^b^	1397.0 ± 163.0 ^c^
*Hydroxycinnamic acid derivatives*							
Chlorogenic acid	120.9 ± 9.9 ^a^	121.6 ± 23.0 ^a^	124.2 ± 14.1 ^a^	104.5 ± 14.4 ^a^	nd	15.9 ± 2.2 ^b^	36.0 ± 5.0 ^b^
Cryptochlorogenic acid (*cis*)	127.4 ± 5.7 ^b^	155.1 ± 19.2 ^a^	137.4 ± 9.5 ^ab^	145.0 ± 4.3 ^ab^	nd	22.0 ± 0.7 ^c^	38.1 ± 1.1 ^c^
Cryptochlorogenic acid (*trans*)	1450.0 ± 3.7 ^a^	1508.2 ± 21.0 ^a^	1137.9 ± 97.0 ^b^	651.6 ± 7.6 ^c^	231.1 ± 8.3 ^de^	67.2 ± 0.8 ^f^	171.2 ± 2.0 ^ef^
Caffeic acid	nd	nd	nd	nd	1684.2 ± 201.7 ^a^	–	–
5-feruloylquinic acid	86.7 ± 1.4 ^ab^	81.1 ± 4.0 ^b^	96.8 ± 7.8 ^a^	nd	nd	–	–
*p*-coumaric acid	nd	nd	nd	nd	22.6 ± 2.1 ^a^	–	–
3,5-dicaffeoylquinic acid	70.3 ± 1.3 ^a^	73.5 ± 8.1 ^a^	67.4 ± 4.7 ^a^	68.7 ± 7.9 ^a^	t	7.0 ± 0.8 ^c^	23.5 ± 2.7 ^b^
5-*p*-coumaroylquinic acid	t	t	t	t	t	–	–
*Flavones*							
Vicenin-2	80.3 ± 11.3 ^a^	70.3 ± 10.9 ^a^	69.5 ± 8.3 ^a^	nd	nd	–	–
*Flavonols*							
Rutin	145.4 ± 2.0 ^a^	148.7 ± 8.7 ^a^	138.2 ± 12.8 ^a^	nd	t	–	–
Isoquercetin	89.4 ± 2.0 ^ab^	98.8 ± 6.5 ^a^	85.4 ± 8.3 ^b^	t	t	–	–
*Methylxanthines*							
Caffeine	7879.5 ± 343.4 ^abc^	8339.0 ± 298.3 ^a^	8137.1 ± 852.2 ^ab^	6689.4 ± 700.9 ^bcd^	6539.4 ± 932.8 ^cd^	6549.0 ± 686.1 ^cd^	5959.3 ± 624.4 ^d^

Results are reported as mean ± SD (*n* = 3). Mean values within rows followed by different superscript letters (a, b, c, d, e, f) are significantly different when subjected to Tukey’s test (*p* < 0.05). ND: Non-Digested; OP: oral phase; GP: gastric phase; IP: intestinal phase; CP: colonic phase; C2A: Caco-2 absorption; HIA: human intestinal absorption; nd: non-detected; t: traces.

**Table 3 antioxidants-11-01818-t003:** Retention index, bioaccessibility, and potential bioavailability (%) of individual phenolic compounds and methylxanthines from the coffee pulp flour (CPF) and extract (CPE) throughout simulated gastrointestinal digestion.

Compounds	Retention Index		Bioaccessibility		Bioavailability	
	OP	GP	IP	CP	C2A	HIA
** *Coffee pulp flour* **						
*Hydroxybenzoic acid derivatives*						
Gallic acid	60.1 ± 7.7 ^a^**	53.2 ± 4.8 ^a^*	30.6 ± 5.6 ^b^	22.1 ± 5.6 ^bc^***	16.5 ± 3.0 ^c^	13.2 ± 2.4 ^c^
Protocatechuic acid	61.4 ± 1.4 ^bc^***	74.9 ± 5.0 ^abc^*	91.9 ± 14.0 ^a^	62.0 ± 6.9 ^bc^	80.3 ± 12.2 ^ab^*	58.0 ± 8.8 ^c^*
*Hydroxycinnamic acid derivatives*						
Chlorogenic acid	56.6 ± 10.3 ^c^*	76.2 ± 18.9 ^bc^	123.2 ± 22.2 ^a^*	109.0 ± 22.8 ^ab^	18.8 ± 3.4 ^d^*	42.4 ± 7.6 ^cd^*
Cryptochlorogenic acid (*cis*)	62.6 ± 7.7 ^cd^**	88.4 ± 14.4 ^bc^	122.7 ± 28.1 ^ab^	142.1 ± 29.9 ^a^	18.6 ± 4.2 ^e^	32.2 ± 7.4 ^de^
Cryptochlorogenic acid (*trans*)	53.8 ± 1.6 ^b^***	74.4 ± 6.2 ^a^	79.3 ± 9.9 ^a^***	21.1 ± 2.8 ^c^*	8.2 ± 1.0 ^c^***	20.8 ± 2.6 ^c^***
Neochlorogenic acid	39.7 ± 5.3 ^b^	33.0 ± 6.6 ^b^	71.2 ± 14.2 ^a^	–	10.8 ± 2.2 ^c^	24.5 ± 4.9 ^bc^
5-feruloylquinic acid	–	68.6 ± 6.8 ^a^***	–	–	–	–
3,5-dicaffeoylquinic acid	29.1 ± 3.6 ^cd^***	45.0 ± 5.1 ^b^***	78.9 ± 10.3 ^a^	41.2 ± 6.3 ^bc^	8.0 ± 1.1 ^e^	27.0 ± 3.5 ^d^
5-*p*-coumaroylquinic acid	6.7 ± 1.0 ^d^	19.7 ± 2.5 ^bc^	81.8 ± 12.0 ^a^	14.7 ± 1.3 ^cd^	20.2 ± 3.0 ^bc^	31.1 ± 4.6 ^b^
*Flavones*						
Vicenin-2	27.8 ± 1.5 ^c^**	53.3 ± 7.1 ^a^*	41.9 ± 6.8 ^b^	31.3 ± 3.5 ^c^	5.0 ± 0.8 ^d^	7.8 ± 3.1 ^d^
*Flavonols*						
Quercetin 3,7-dihexoside	45.4 ± 5.0 ^b^	66.1 ± 3.3 ^a^	62.8 ± 7.7 ^a^	–	6.9 ± 0.8 ^c^	4.7 ± 0.6 ^c^
Rutin	43.0 ± 2.5 ^b^***	69.0 ± 9.4 ^a^**	58.4 ± 7.6 ^a^	–	8.0 ± 1.0 ^c^	13.0 ± 1.7 ^c^
Isoquercetin	30.7 ± 3.5 ^bc^***	50.6 ± 8.8 ^a^***	57.3 ± 11.1 ^a^	34.0 ± 5.4 ^b^	25.3 ± 4.9 ^bc^	18.2 ± 3.5 ^bc^
*Methylxanthines*						
Caffeine	42.6 ± 0.5 ^c^***	63.6 ± 7.5 ^b^***	82.2 ± 5.7 ^a^	76.5 ± 1.6 ^a^	80.4 ± 5.6 ^a^	72.2 ± 5.1 ^ab^
** *Coffee pulp extract* **						
*Hydroxybenzoic acid derivatives*						
Gallic acid	111.0 ± 18.0 ^a^	68.1 ± 8.2 ^b^	32.6 ± 5.3 ^c^	97.4 ± 16.6 ^a^	17.5 ± 2.9 ^c^	14.1 ± 2.3 ^c^
Protocatechuic acid	104.3 ± 6.1 ^a^	97.0 ± 11.3 ^a^	70.6 ± 9.9 ^b^	74.7 ± 11.2 ^b^	61.7 ± 8.7 ^b^	44.6 ± 6.3 ^c^
*Hydroxycinnamic acid derivatives*						
Chlorogenic acid	100.6 ± 27.2 ^a^	102.8 ± 20.1 ^a^	86.5 ± 18.9 ^a^	–	13.2 ± 2.9 ^b^	29.8 ± 6.5 ^b^
Cryptochlorogenic acid (*cis*)	121.7 ± 20.5 ^a^	107.9 ± 12.3 ^a^	113.8 ± 8.5 ^a^	–	17.2 ± 1.3 ^b^	29.9 ± 2.2 ^b^
Cryptochlorogenic acid (*trans*)	104.0 ± 1.7 ^a^	78.5 ± 6.9 ^a^	44.9 ± 0.6 ^c^	15.9 ± 0.6 ^c^	4.6 ± 0.1 ^d^	11.8 ± 0.2 ^cd^
5-feruloylquinic acid	93.5 ± 6.1 ^a^	111.6 ± 10.7 ^a^	–	–	–	–
3,5-dicaffeoylquinic acid	104.6 ± 13.4 ^a^	95.8 ± 8.4 ^a^	97.7 ± 13.1 ^a^	–	10.0 ± 1.3 ^c^	33.4 ± 4.5 ^b^
*Flavones*						
Vicenin-2	87.5 ± 25.8 ^a^	86.5 ± 22.5 ^a^	–	–	–	–
*Flavonols*						
Rutin	102.3 ± 7.4 ^a^	95.1 ± 10.1 ^a^	–	–	–	–
Isoquercetin	110.6 ± 9.7 ^a^	95.6 ± 11.4 ^a^	–	–	–	–
*Methylxanthines*						
Caffeine	105.8 ± 8.4 ^a^	103.3 ± 15.3 ^ab^	84.9 ± 12.6 ^ab^	83.0 ± 15.5 ^ab^	83.1 ± 12.3 ^ab^	75.6 ± 11.2 ^b^

Results are reported as mean ± SD (*n* = 3). Mean values within a line followed by different superscript letters (a, b, c, d, e) are significantly different when subjected to Tukey’s test (*p* < 0.05). OP: oral phase; GP: gastric phase; IP: intestinal phase; CP: colonic phase; C2A: Caco-2 absorption; HIA: human intestinal absorption; nd: non-detected; t: traces. Mean values followed by superscript asterisks significantly differ (CPF vs. CPE) when subjected to *T*-test (* *p* < 0.05, ** *p* < 0.01, *** *p* < 0.001).

## Data Availability

Data is contained within the article or Appendix A.

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
