# Peer review of "Understanding the Gastrointestinal Behavior of the Coffee Pulp Phenolic Compounds under Simulated Conditions"

_antioxidants, 2022, doi:10.3390/antiox11091818_

Round 1

Reviewer 1 Report

The manuscript concerns the characteristics of coffee fruit pulp, the content of phenolic components and their bioavailability determined in vitro and partly in silico. The research brings new knowledge, it is well planned and described. Therefore, it should be published after a few corrections:

Page 2, section 2. Materials and Methods: The chemicals used in the tests are not described

Page 2, section 2. Materials and Methods: if the methods used are not own methods, relevant references should be added.

Page 3, subsection 2.2. In vitro simulated gastrointestinal digestion: reference should be added to prove that the coffee fruit does not contain starch.

Page 4, line 72-73: for hydroxybenzoic, hydroxycinnamic acids, and caffeine the preferred wavelengths for DAD detection was 280 nm: add the reference for a confirmation that this is the maximum absorbance of the UV spectrum of hydroxycinnamic acids, as the most commonly used wavelength of the absorbance maximum is 325 nm .

Page 12, subsection 3.4. Caffeine and protocatechuic acid were the main compounds absorbed in the intestine after gastrointestinal digestion: caffeine accounted for 58.7% of CPF; does this mean some restrictions on CPF consumption? As it is known, caffeine consumed in higher amounts may be toxic, about 10 g of caffeine is a lethal dose for an adult.

Page 16, line 600-608: the results of Grzelczyk et al. should be discussed (Joanna Grzelczyk, Dominik Szwajgier, Ewa Baranowska-Wójcik, Grażyna Budryn, MaÅ‚gorzata ZakÅ‚os-Szyda, Bożena Sosnowska, Bioaccessibility of coffee bean hydroxycinnamic acids during in vitro digestion influenced by the degree of roasting and activity of intestinal probiotic bacteria, and their activity in Caco-2 and HT29 cells, Food Chemistry, Volume 392, 2022, 133328, doi: org / 10.1016 / j.foodchem.2022.133328.

Author Response

The Author's Reply to the Review Report can be found in the attached document

Reviewer 2 Report

The manuscript entitled "Understanding the gastrointestinal behavior of the coffee pulp phenolic compounds under simulated conditions" has been revised and found it a promising and outstanding research and might be brings a new understanding about the gastrointestinal behavior of the coffee pulp phenolic compounds and caffeine under simulated conditions. The manuscript is well organized and the results are well discussed and interpreted. So, the manuscript is appropriate for considering for publication in Antioxidants after considering the following minor comments:-

1. in the section materials and methods- All instruments which used in the experiments must be identified (Model, city and country must be presented)

2. All chemicals and reagents must be characterized the the source of chemical and reagents must be added 

3. In the line 94-95 " the CPF (0.02 g mL−1 solid-to-solvent ratio) was added to boiling water (100 °C) and mixed for 90 min.". This high temperature for a 90 min doesn't affect the phenolic compounds and destroy the phenolic compound's structure?

4. The conclusion must be improved to focus on the novelty and new findings, which must be supported by the results. 

Author Response

(The authors gave the same response as above.)
